# High-brightness scalable continuous-wave single-mode photonic-crystal laser

Masahiro Yoshida[1,3], Shumpei Katsuno[1,3], Takuya Inoue[2,3], John Gelleta[1], Koki Izumi[1], Menaka De Zoysa[2], Kenji Ishizaki[2] & Susumu Noda[1,2 ✉]

Realizing large-scale single-mode, high-power, high-beam-quality semiconductor lasers, which rival (or even replace) bulky gas and solid-state lasers, is one of the ultimate goals of photonics and laser physics. Conventional high-power semiconductor lasers, however, inevitably suffer from poor beam quality owing to the onset of many-mode oscillation[1,2], and, moreover, the oscillation is destabilized by disruptive thermal effects under continuous-wave (CW) operation[3,4]. Here, we surmount these challenges by developing large-scale photonic-crystal surface-emitting lasers with controlled Hermitian and non-Hermitian couplings inside the photonic crystal and a pre-installed spatial distribution of the lattice constant, which maintains these couplings even under CW conditions. A CW output power exceeding 50 W with purely single-mode oscillation and an exceptionally narrow beam divergence of 0.05° has been achieved for photonic-crystal surface-emitting lasers with a large resonant diameter of 3 mm, corresponding to over 10,000 wavelengths in the material. The brightness, a figure of merit encapsulating both output power and beam quality, reaches 1 GW cm$^{-2}$ sr$^{-1}$, which rivals those of existing bulky lasers. Our work is an important milestone toward the advent of single-mode 1-kW-class semiconductor lasers, which are expected to replace conventional, bulkier lasers in the near future.

Semiconductor lasers boast various beneficial features that cannot be found in other lasers (for example, gas, solid-state and fibre lasers), such as compactness, high efficiency and high controllability, and they are key devices for various applications in modern society, including telecommunications and optical recording. Realizing semiconductor lasers that also operate in a single mode with high output power and high beam quality remains an ultimate yet elusive goal in photonics and laser physics. Such semiconductor lasers are in demand for many emerging applications, including next-generation laser processing, remote sensing, long-range free-space communications and even light propulsion for spaceflight[5–8]. Conventional semiconductor lasers are limited by the maximum emission area that can support single-mode operation; namely, widening the emission area to increase the output power leads to the onset of many-mode oscillation, which degrades the beam quality[1,2]. Even worse, under continuous-wave (CW) operation, the beam quality is prone to degrade further due to a thermally induced refractive index distribution inside the resonator, which is one of the critical factors responsible for unstable oscillation[3,4] (Supplementary Text Section 1 has more details).

The photonic-crystal surface-emitting laser (PCSEL)[9–15] shows potential to overcome the above limitations of conventional semiconductor lasers. The PCSEL achieves lasing oscillation of a two-dimensional standing wave at a singularity (Γ, M and so on) point in its photonic band structure. By tailoring the design of the unit cell of its photonic crystal, the mutual optical couplings inside the photonic crystal can be tuned to enable single-mode oscillation over a large area. One unit-cell design

proposed for this purpose is the double lattice[15], in which one lattice point group is shifted from a second in the $x$ and $y$ directions by approximately one quarter of the wavelength in the material. In this double lattice, the strength of in-plane optical coupling, which can be referred to as Hermitian coupling since it is not accompanied by radiation loss, is weakened by destructive interference of waves diffracted by 180° and 90° at each of the two lattice points. Consequently, optical losses of higher-order modes from the periphery of the resonator increase compared with that of the fundamental mode, resulting in a wider threshold gain margin between these modes and thus, more stable oscillation in the fundamental mode. Based on this concept, CW lasing oscillation with an output power of 7 W and a brightness of 180 MW cm$^{-2}$ sr$^{-1}$ was experimentally demonstrated using PCSELs with circular resonant diameters of 800 μm (ref. 15).

Following the above developments, a design guideline to realize single-mode oscillation over areas of even larger (≥3 mm) diameters was recently reported[16] based on the control of not only the Hermitian coupling described above, but also non-Hermitian coupling, which accompanies radiation loss. In addition, another related approach toward realizing scalable single-mode photonic-crystal lasers was also reported[17], although the resonant (emission) size achieved in experiments therein was less than approximately 64 μm. (Supplementary Text Section 2 has a brief comparison between these two approaches).

The most important outstanding challenges are, therefore, twofold. One is to investigate whether single-mode operation of PCSELs can be truly scaled to extremely large (≥3 mm) diameters; the other is to investigate whether single-mode operation can be maintained even under

[1]Department of Electronic Science and Engineering, Kyoto University, Kyoto, Japan. [2]Photonics and Electronics Science and Engineering Center, Kyoto University, Kyoto, Japan. [3]These authors contributed equally: Masahiro Yoshida, Shumpei Katsuno, Takuya Inoue. ✉e-mail: snoda@kuee.kyoto-u.ac.jp

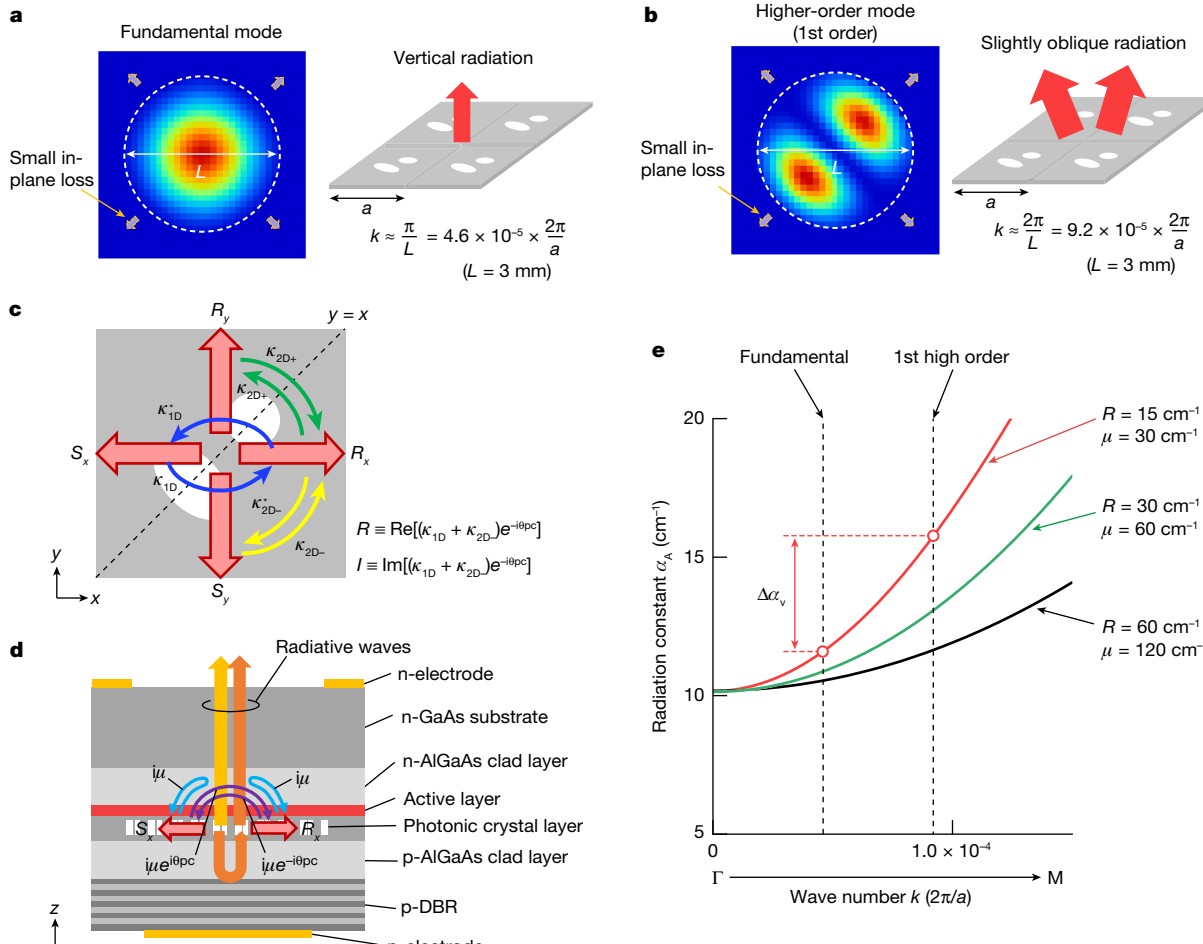

**Fig. 1 | Control of Hermitian and non-Hermitian optical couplings inside the photonic crystal to increase the threshold gain margin between the fundamental and higher-order modes. a**, Typical electric-field intensity distribution (left panel) and schematic of radiation (right panel) of the fundamental mode in a PCSEL with a diameter of $L$. **b**, Typical electric-field intensity distribution (left panel) and schematic of radiation (right panel) of a higher-order mode in the same PCSEL. The higher-order mode emits a multilobed beam in a slightly oblique direction. **c**, Hermitian couplings between the four fundamental waves ($R_x$, $S_x$, $R_y$ and $S_y$) inside a PCSEL with a double-lattice photonic crystal. **d**, Schematic of the cross-sectional structure of the PCSEL and the non-Hermitian couplings between the fundamental waves via radiative waves inside the device, where the couplings between $R_x$ and $S_x$ are illustrated as an example. A backside reflector (p-type distributed Bragg reflector (p-DBR)) is used to control the magnitude of the non-Hermitian coupling coefficient. **e**, Calculated radiation constants of mode A as functions of $k$ when the values of $R$ and $\mu$ are changed.

CW conditions, where disruptive thermal effects appear. Here, we first show that it is indeed possible to realize single-mode oscillation even in a PCSEL with a 3-mm diameter, corresponding to over 10,000 wavelengths in the material, by simultaneously controlling the Hermitian and non-Hermitian couplings. Then, we introduce a lattice-constant distribution to compensate for the thermal effects and thereby maintain the controlled Hermitian and non-Hermitian couplings even under CW conditions. By doing so, we finally experimentally achieve 50-W CW operation in a single mode (with a single wavelength) with a very narrow beam divergence angle of 0.05° (corresponding to a beam quality $M^2 \approx 2.36$). The brightness of the developed laser reaches 1 GW cm$^{-2}$ sr$^{-1}$, which is more than one order of magnitude greater than that of conventional semiconductor lasers and even rivals those of existing bulky gas and solid-state lasers. The strategies demonstrated here are expected to be applicable to scaling up the diameter of the device to 10 mm, leading to the 1-kW class, high-beam-quality operation of PCSELs.

## Hermitian and non-Hermitian-controlled PCSEL

First, we describe the strategy to realize single-mode oscillation in a large-scale PCSEL. The left panels of Fig. 1a,b show typical electric-field

distributions of the fundamental and higher-order modes in a PCSEL with a diameter of $L$. As $L$ increases, the in-plane losses (that is, light escaping from the periphery of the resonator) of both the fundamental and higher-order modes converge toward zero, and thus, the ability to discriminate between these modes via in-plane loss greatly diminishes. Accordingly, we instead consider mode discrimination via vertical radiation loss (that is, the radiation constant), which remains high even when $L$ becomes large. As illustrated in the right panels of Fig. 1a,b, the (first) higher-order mode is double lobed, and consequently, it has a slightly larger in-plane wave number than the single-lobed fundamental mode. Our strategy is to make the radiation constant at the wave number corresponding to the (first) higher-order mode sufficiently larger than that of the fundamental mode by controlling the Hermitian and non-Hermitian couplings in a double-lattice photonic crystal.

Figure 1c,d illustrates the Hermitian and non-Hermitian couplings of four fundamental waves, labelled $R_x$, $S_x$, $R_y$ and $S_y$, in a double-lattice structure, where the former and latter couplings do not and do accompany radiation loss, respectively. In Fig. 1c, the coefficients $\kappa_{1D}$ and $\kappa_{2D\pm}$ express the strengths of Hermitian coupling at angles of 180° and ±90°, respectively. The self-coupling of the four fundamental waves without radiation loss is expressed by $\kappa_{11}$ (not shown in the figure). In Fig. 1d,

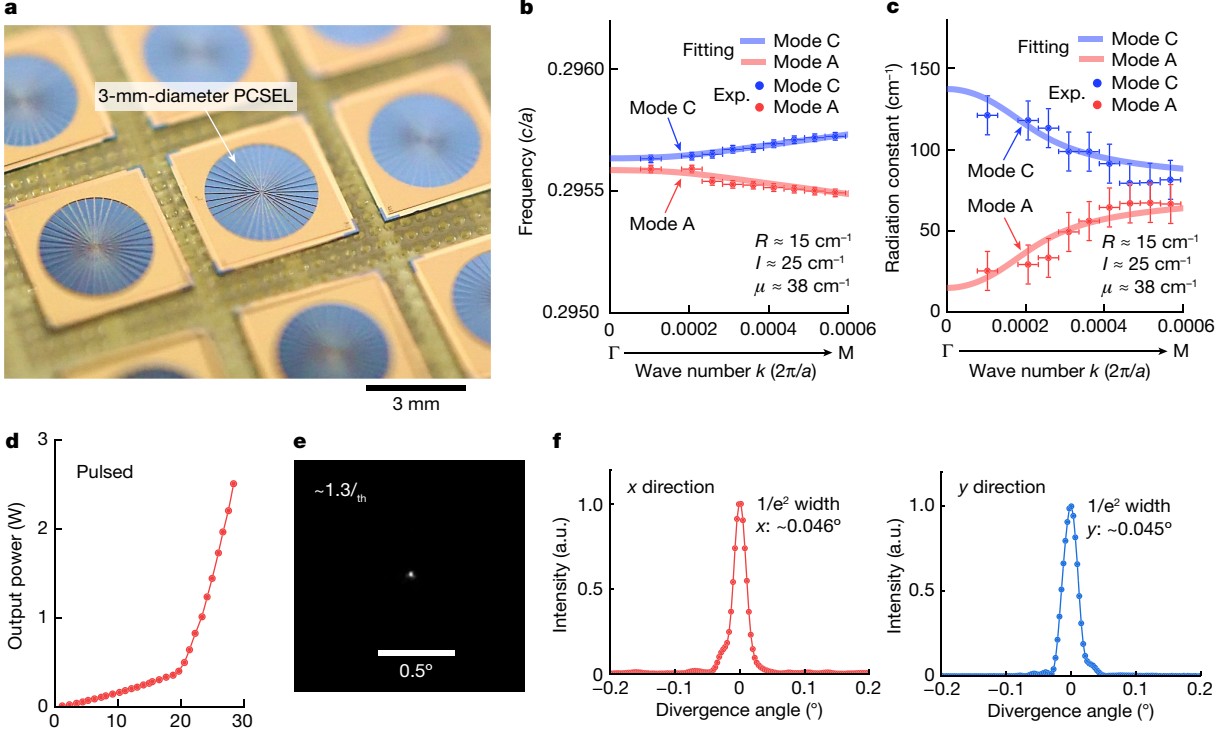

**Fig. 2 | Development of a 3-mm-diameter PCSEL and demonstration of very narrow beam divergence under pulsed operation. a**, Photograph of the fabricated 3-mm-diameter PCSEL chips. **b**,**c**, Frequencies (**b**) and radiation constants (**c**) as functions of $k$ for modes A and C. The dots indicate experimental data (Exp.) obtained from the peak wavelengths and line widths of the spontaneous emission spectrum at each wave number. The solid lines are theoretical curves based on equation (1), where the best-fit results are $R = 15$ cm$^{-1}$, $I = 25$ cm$^{-1}$ and $\mu = 38$ cm$^{-1}$. The vertical error bars in **b** and **c** represent measurement uncertainty associated with the central frequency and

the spectral line width, and the horizontal error bars represent measurement uncertainty associated with the wave number. **d**, Measured $I$–$L$ characteristics under pulsed conditions. The pulse width and repetition frequency were set to 200 ns and 200 Hz, respectively. **e**, Measured FFP at an injection current of $1.3I_{th}$. **f**, Cross-sectional intensity profiles of the FFP along the $x$ and $y$ directions. A very narrow $1/e^2$ divergence angle of 0.045° was obtained owing to the achievement of small yet appropriately balanced values of $R$ and $\mu$ as described in the text. a.u., arbitrary unit.

the coefficients i$\mu$ and i$\mu$e$^{\pm i\theta_{pc}}$ express the strengths of non-Hermitian couplings at angles of 0° (that is, self-coupling) and 180°, respectively, via the radiative waves, where $\mu$ is the magnitude of the non-Hermitian coupling coefficient, $\theta_{pc}$ is a phase change associated with ±180° non-Hermitian coupling and i is the imaginary unit.

Owing to such optical couplings among the four fundamental waves, four resonant cavity modes are constructed. In a double-lattice photonic crystal with mirror symmetry about the axis of $y = x$, two of these modes (labelled A and C) are antisymmetric modes whose electric-field vectors are antisymmetric about this axis, and the remaining two modes (labelled B and D) are symmetric modes whose electric-field vectors are symmetric about this axis. The radiation constant, as well as the frequency, of each mode is analytically derived as a function of the in-plane wave number $k$ using three-dimensional coupled-wave theory[16,18]. The frequency and radiation constant ($\delta_A$, $\alpha_A$) of mode A, which has the lowest radiation constant among all four modes, and those ($\delta_C$, $\alpha_C$) of its counterpart mode C are expressed as (Methods has the details of the derivation)

$$\delta_{A,C} + i\frac{\alpha_{A,C}}{2} = \kappa_{11} + \kappa_{2D+} + i\mu \mp \sqrt{[R + iI + i\mu][R - iI + i\mu] + \left(\frac{k}{\sqrt{2}}\right)^2}. \quad (1)$$

Here, $\delta_A + i\alpha_A/2$ and $\delta_C + i\alpha_C/2$ correspond to the negative and positive square-root terms, respectively, and the in-plane wave number $k$ is taken along the Γ–M direction (that is, $k_x = k_y = k/\sqrt{2}$), so that the symmetry of the electric fields coincides with the symmetry of the photonic crystal along the line of $y = x$. In equation (1), $R$ and $I$ correspond to the

real and imaginary parts of the Hermitian coupling coefficient, respectively; namely, $R \equiv \mathrm{Re}[(\kappa_{1D} + \kappa_{2D-})e^{-i\theta_{pc}}]$ and $I \equiv \mathrm{Im}[(\kappa_{1D} + \kappa_{2D-})e^{-i\theta_{pc}}]$. $R$ expresses the overall strength of in-plane feedback of combined 180° and 90° diffractions at the Γ point, which determines the size of the frequency gap between modes A and C. $I$ determines the phase of the in-plane electric fields, specifically the position of the electric-field node with respect to the position of the air holes, and consequently, it determines the degree of cancellation of the vertical radiation in mode A at the Γ point. (Details on $R$ and $I$ are explained in Supplementary Text Section 3).

Figure 1e shows $\alpha_A$ as a function of $k$ calculated using equation (1) for several selections of $R$ and $\mu$, while $I$ was adjusted so that $\alpha_A$ at $k = 0$ was identical in all cases. Evidently, decreasing $R$ and $\mu$ simultaneously while maintaining their balance increases the curvature of dispersion around the Γ point, resulting in an abrupt change of $\alpha_A$ with respect to $k$. Consequently, the threshold margin $\Delta\alpha_v$ between the fundamental and higher-order modes near the Γ point can be increased, whereupon single-mode oscillation is expected even for large 3-mm-diameter PCSELs (Methods has a note on $\Delta\alpha_v$).

Based on the above strategy, we developed a 3-mm-diameter PCSEL, whose $R$ and $I$ values were controlled by changing the lattice separation ($d$) and the balance of air-hole sizes ($x$) of a double-lattice structure (Fig. 1c and Supplementary Text Section 4) and whose $\mu$ value was controlled by changing the thickness of the p-AlGaAs clad layer, which affects the degree of optical interference between front side-emitted and back side-reflected radiative waves[16] (Fig. 1d and Supplementary Text Section 4). Figure 2a shows finished 3-mm-diameter PCSELs

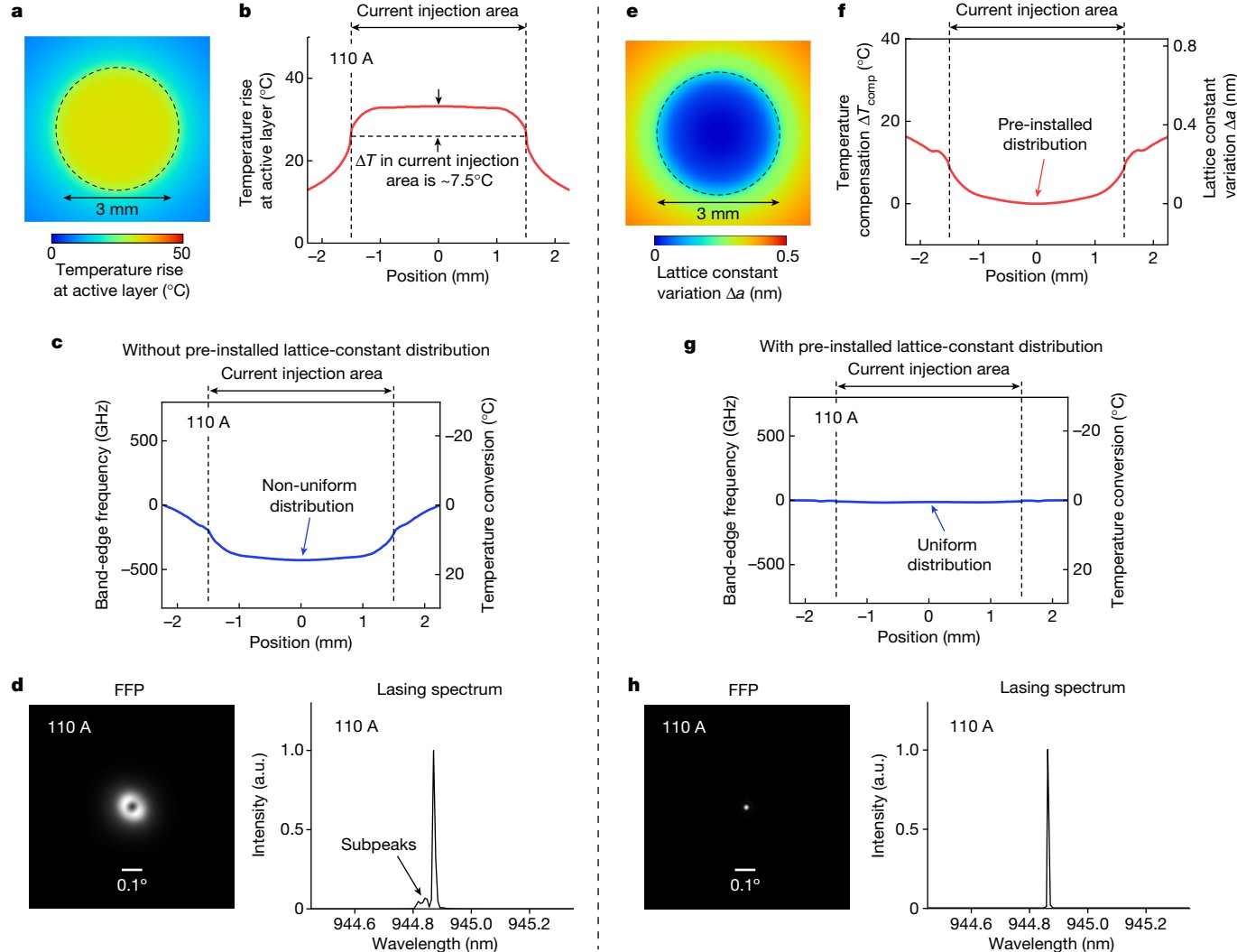

**Fig. 3 | Introduction of a lattice-constant distribution to realize single-mode oscillation of a 3-mm-diameter PCSEL even under CW operation. a,b,** Calculated in-plane distribution (**a**) and its profile (**b**) of temperature rise near the active layer at a CW injection current of 110 A. From these calculations, the temperature difference $\Delta T$ between the centre and the edge of the current injection area was estimated to be 7.5 °C. **c,** Band-edge frequency distribution of a 3-mm-diameter PCSEL without a pre-installed lattice-constant distribution. The right axis shows the temperature change converted from the frequency. A non-uniform band-edge frequency distribution is formed due to the temperature-induced refractive index distribution. **d,** FFP and lasing spectrum of a 3-mm-diameter PCSEL without a

pre-installed lattice-constant distribution. **e,** In-plane distribution of the lattice-constant variation to compensate for the temperature distribution that appears at an injection current of 110 A. **f,** Lattice-constant variation and corresponding temperature compensation as functions of position. **g,** Band-edge frequency distribution of a 3-mm-diameter PCSEL with the pre-installed lattice-constant distribution. The right axis shows the temperature change converted from the frequency. A uniform frequency distribution is realized by the mutual cancellation of the temperature-induced and lattice constant-induced changes to the band-edge frequency. **h,** FFP and lasing spectrum of a 3-mm-diameter PCSEL with the pre-installed lattice-constant distribution.

fabricated based on an air hole-retained metal–organic vapour-phase epitaxy regrowth technique[19]. A mesh-window n-electrode was deposited onto an n-GaAs substrate (emission side) for uniform current injection across the entire 3-mm-diameter area.

We first measured the frequencies and radiation constants of modes A and C around the Γ point of the fabricated device as plotted in Fig. 2b,c and then, estimated the coupling coefficients $R$, $I$ and $\mu$ by fitting the analytical values given by equation (1) to their experimental values (Methods has details). The best-fit results were $R \approx 15$ cm$^{-1}$, $I \approx 25$ cm$^{-1}$ and $\mu \approx 38$ cm$^{-1}$; these values correspond to the case of the red line in Fig. 1e, in which a large threshold margin $\Delta\alpha_v$ is expected. Then, we measured lasing characteristics of the PCSEL under pulsed conditions. Here, the PCSEL was not mounted to a heat sink, which limited the maximum tolerable injection current and hence output power even under pulsed conditions. The current–light output ($I$–$L$) characteristics measured at

room temperature in Fig. 2d show that lasing oscillation occurred at a threshold injection current of $I_{th} \approx 20$ A. Figure 2e shows a far-field pattern (FFP) of the emitted beam above the threshold ($1.3 I_{th}$). As indicated by the cross-sectional profile of the FFP (Fig. 2f), a very narrow beam divergence angle of 0.045° was achieved, which we attribute to the small yet appropriately balanced values of $R$ and $\mu$ as described above.

## High-brightness CW single-mode PCSEL

CW current injection induces a spatially non-uniform temperature distribution inside the PCSEL due to the accumulation of heat. We simulated the effects of heat accumulation on the lasing characteristics of a 3-mm-diameter PCSEL under CW conditions based on a self-consistent analysis[20] of the interaction among photons, carriers and thermal effects (Supplementary Text Section 5 for details).

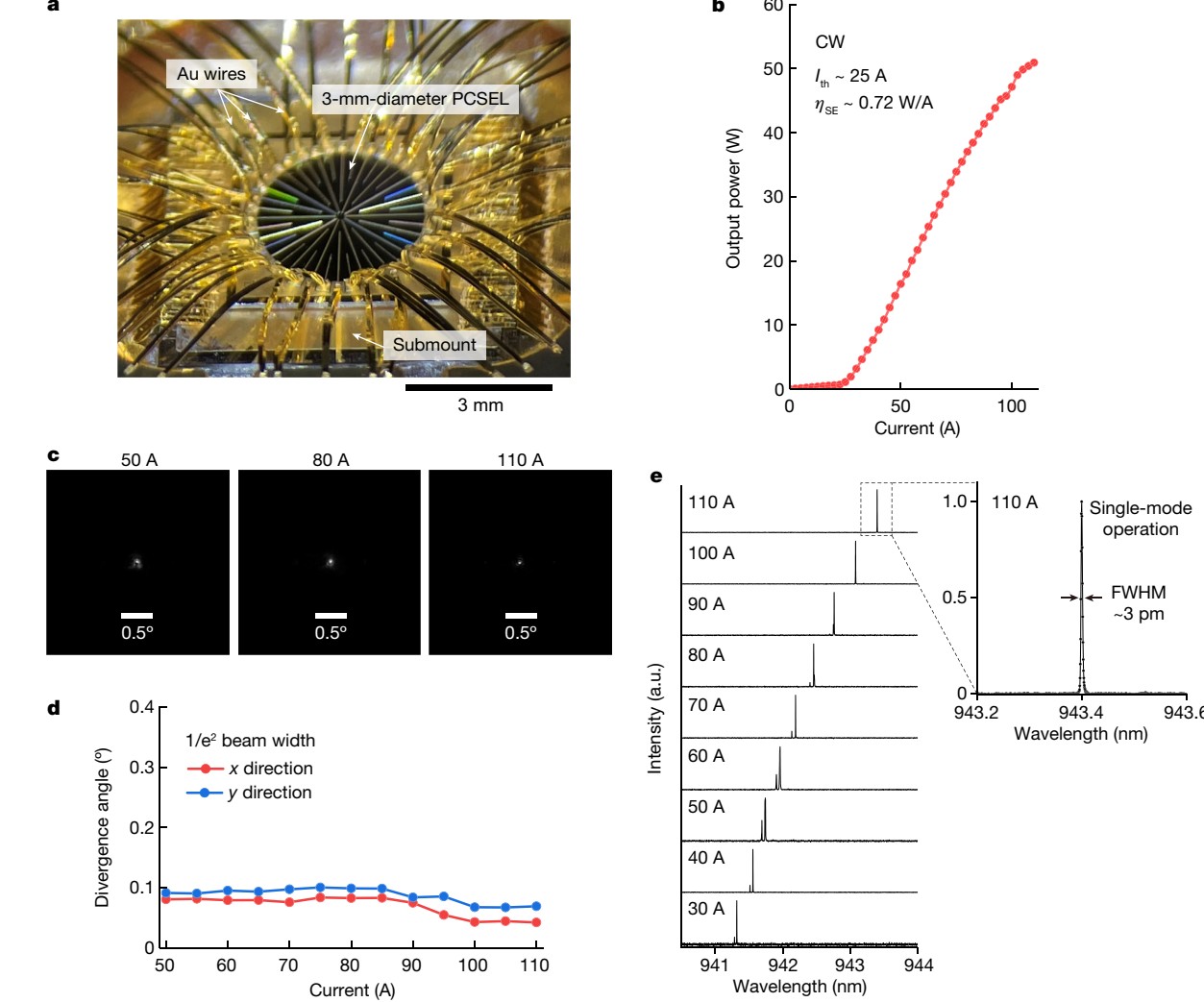

**Fig. 4 | Demonstration of 50-W single-mode CW operation of a 3-mm-diameter PCSEL. a**, Photograph of the 3-mm-diameter PCSEL, which was mounted on a package via a submount with high thermal conductivity in the p-side-down configuration. To inject a high CW current, many Au wires were bonded to the n-side electrode of the PCSEL and to the submount surface (p side). **b**, $I$–$L$ characteristics under CW conditions, where the heat sink was cooled by water at a temperature of 6 °C during measurements. **c**, FFPs at injection currents of 50, 80 and 110 A. **d**, The $1/e^2$ beam divergence angle versus injection current along the $x$ and $y$ directions. Very narrow beam divergence angles of 0.05° were obtained. **e**, Lasing spectra measured at various injection currents. A narrow spectral width was observed. Single-mode oscillation was realized at injection currents of 100–110 A. FWHM, full width at half maximum.

Figure 3a,b shows the calculated in-plane temperature distribution near the active layer at a sufficiently large CW injection current of 110 A (chosen upon consideration of the experimental conditions described later). As shown, the temperature at the centre of the current injection area becomes higher than at the periphery, which results in the downward convex-shaped band-edge frequency distribution shown in Fig. 3c via a change in refractive index. This frequency distribution perturbs the electric field of the fundamental mode, and consequently, it induces multimodal behaviour and broadens the emitted beam, as shown in Fig. 3d. To suppress such unwanted effects, we introduce a spatial variation $\Delta a(x,y)$ to the lattice constant of the photonic crystal, which compensates for a temperature distribution $\Delta T_{comp}(x,y)$, as shown in Fig. 3e,f (Methods has details). Figure 3g shows the band-edge frequency distribution calculated at an injection current of 110 A following the introduction of this lattice-constant distribution. This figure clearly shows that a uniform frequency distribution is obtained, owing to the mutual cancellation of the temperature-induced and lattice constant-induced changes of the band-edge frequency. As a result, the emission of a single-mode beam with a very narrow divergence angle is expected to be obtained, as shown in Fig. 3h.

Applying the above strategy, we developed the 3-mm-diameter PCSEL with the pre-installed lattice-constant distribution. Figure 4a shows a finished 3-mm-diameter PCSEL mounted on a package. The coupling coefficients of the device were estimated as $R \approx 24$ cm$^{-1}$, $I \approx 14$ cm$^{-1}$ and $\mu \approx 44$ cm$^{-1}$ (Methods has details). Figure 4b shows $I$–$L$ characteristics of the 3-mm-diameter PCSEL under CW conditions. The threshold current was 25 A, and the slope efficiency was approximately 0.72 W A$^{-1}$. A CW output power exceeding 50 W was obtained from the single-chip PCSEL at injection currents of 100–110 A. Figure 4c shows the FFPs at several injection currents, and Fig. 4d shows the $1/e^2$ beamwidths of FFPs as functions of the injection current. Remarkably, the divergence angles in the $x$ and $y$ directions became minimal (0.05°) at 100–110 A, where the frequency distributions due to the temperature-induced refractive index change and the pre-installed lattice-constant distribution were designed to cancel each other out. We note that the divergence angle was slightly larger in the $y$ direction than in the $x$ direction due to a small residual side lobe with an intensity of approximately $1/e^2$ of that of the main peak; this side lobe can be eliminated in the future by further optimizing the pre-installed lattice-constant distribution. From the divergence angles, including the small residual side lobe, the beam quality

$M^2$ was estimated to be 2.36. The CW laser brightness, evaluated using the measured output power and FFP widths at 110 A, was 1 GW cm$^{-2}$ sr$^{-1}$. Furthermore, the laser spectra at several injection currents are shown in Fig. 4e. Although several modes were seen to oscillate at lower injection currents, single-mode oscillation was achieved at injection currents of 100–110 A (corresponding to a CW output power of around 50 W). The measured spectral width at this injection current was 3 pm, which was limited by the spectral resolution of our spectrometer. As this resolution was finer than the predicted spectral spacing between the fundamental and next higher-order modes, we may say that purely single-mode oscillation was achieved. We note that $M^2 \geq 2$ in spite of single-mode oscillation is due to the super-Gaussian electromagnetic-field intensity profile caused predominantly by the uniform current injection. The dependence of the laser spectrum on the injection current agrees with that of the beam divergence angle plotted in Fig. 4d. These results show that the pre-installed lattice-constant distribution together with the control of Hermitian and non-Hermitian couplings inside the double-lattice structure has contributed to the realization of purely single-mode, high beam quality, high-power CW operation of an ultra-large-area PCSEL.

## Conclusions

We have developed large-scale PCSELs by control of Hermitian and non-Hermitian couplings to suppress the oscillation of higher-order modes, and we have introduced a lattice-constant distribution to maintain these controlled couplings even under CW operation. By doing so, we have realized 50-W single-mode (or single-wavelength) oscillation of a PCSEL with an ultra-large diameter of 3 mm, corresponding to over 10,000 wavelengths in the material. The 50-W CW output power and a very narrow beam divergence of 0.05° ($M^2 \approx 2.36$) correspond to a brightness of 1 GW cm$^{-2}$ sr$^{-1}$, which rivals those of existing bulky lasers. Controlling the Hermitian and non-Hermitian coupling coefficients ($R$, $I$ and $\mu$) and introducing a lattice-constant distribution suitable for devices of even larger scales (for example, 10-mm diameters) are expected to contribute to the realization of 1-kW class, high-beam-quality operation of PCSELs. Our work is an important milestone toward the replacement of conventional, bulkier solutions and toward innovation in a wide variety of industrial applications, from smart material processing to aerospace applications.

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

# Methods

## Derivation of frequencies and radiation constants of modes A and C

The frequency and radiation constant $(\delta_A, \alpha_A)$ of mode A, which has the lowest radiation constant among all four modes, and those $(\delta_C, \alpha_C)$ of its counterpart mode C can be obtained by solving the following equation based on three-dimensional coupled-wave theory[16,18]:

$$
\left(\delta + i\frac{\alpha}{2}\right)\begin{pmatrix} R_x \\ S_x \\ R_y \\ S_y \end{pmatrix} = \left[\begin{pmatrix} \kappa_{11} & \kappa_{1D} & \kappa_{2D+} & \kappa_{2D-} \\ \kappa_{1D}^* & \kappa_{11} & \kappa_{2D-}^* & \kappa_{2D+} \\ \kappa_{2D+} & \kappa_{2D-} & \kappa_{11} & \kappa_{1D} \\ \kappa_{2D-}^* & \kappa_{2D+} & \kappa_{1D}^* & \kappa_{11} \end{pmatrix}\right.
$$
$$
+ \begin{pmatrix} i\mu & i\mu e^{i\theta_{pc}} & 0 & 0 \\ i\mu e^{-i\theta_{pc}} & i\mu & 0 & 0 \\ 0 & 0 & i\mu & i\mu e^{i\theta_{pc}} \\ 0 & 0 & i\mu e^{-i\theta_{pc}} & i\mu \end{pmatrix} \qquad (2)
$$
$$
\left.+ \begin{pmatrix} k_x & 0 & 0 & 0 \\ 0 & -k_x & 0 & 0 \\ 0 & 0 & k_y & 0 \\ 0 & 0 & 0 & -k_y \end{pmatrix}\right]\begin{pmatrix} R_x \\ S_x \\ R_y \\ S_y \end{pmatrix}.
$$

The first and second terms on the right-hand side of equation (2) represent the Hermitian and non-Hermitian coupling processes described in Fig. 1c,d, respectively. The third term on the right-hand side of equation (2) represents the deviation of the wave number from the Γ point in an arbitrary direction represented by wave numbers $k_x$ and $k_y$, which induces a change in the eigenfrequency of each mode.

Here, we consider the eigenfrequencies of the modes in the Γ−M direction ($k_x = k_y = k/\sqrt{2}$), which is parallel to the axis of symmetry of the double-lattice photonic crystal ($y = x$). Based on the symmetry along $y = x$, the coupled-wave matrices on the right-hand side of equation (2) can be block diagonalized using the basis-transformation matrix $P$:

$$
P = \frac{1}{\sqrt{2}}\begin{pmatrix} 1 & 0 & 1 & 0 \\ 0 & 1 & 0 & 1 \\ 1 & 0 & -1 & 0 \\ 0 & 1 & 0 & -1 \end{pmatrix}, \qquad (3)
$$

as

$$
P^{-1}CP = \begin{pmatrix} \kappa_{11}+\kappa_{2D+} & \kappa_{1D}+\kappa_{2D-} & 0 & 0 \\ \kappa_{1D}^*+\kappa_{2D-}^* & \kappa_{11}+\kappa_{2D+} & 0 & 0 \\ 0 & 0 & \kappa_{11}-\kappa_{2D+} & \kappa_{1D}-\kappa_{2D-} \\ 0 & 0 & \kappa_{1D}^*-\kappa_{2D-}^* & \kappa_{11}-\kappa_{2D+} \end{pmatrix}
$$
$$
+ \begin{pmatrix} i\mu & i\mu e^{i\theta_{pc}} & 0 & 0 \\ i\mu e^{-i\theta_{pc}} & i\mu & 0 & 0 \\ 0 & 0 & i\mu & i\mu e^{i\theta_{pc}} \\ 0 & 0 & i\mu e^{-i\theta_{pc}} & i\mu \end{pmatrix} \qquad (4)
$$
$$
+ \frac{1}{\sqrt{2}}\begin{pmatrix} k & 0 & 0 & 0 \\ 0 & -k & 0 & 0 \\ 0 & 0 & k & 0 \\ 0 & 0 & 0 & -k \end{pmatrix}.
$$

Then, the coupled-wave equation (2) can be divided into the following two forms:

$$
\left(\delta + i\frac{\alpha}{2}\right)\begin{pmatrix} R_x+R_y \\ S_x+S_y \end{pmatrix} = \left[\begin{pmatrix} \kappa_{11}+\kappa_{2D+} & \kappa_{1D}+\kappa_{2D-} \\ \kappa_{1D}^*+\kappa_{2D-}^* & \kappa_{11}+\kappa_{2D+} \end{pmatrix} + \begin{pmatrix} i\mu & i\mu e^{i\theta_{pc}} \\ i\mu e^{-i\theta_{pc}} & i\mu \end{pmatrix}\right.
$$
$$
\left.+ \frac{1}{\sqrt{2}}\begin{pmatrix} k & 0 \\ 0 & -k \end{pmatrix}\right]\begin{pmatrix} R_x+R_y \\ S_x+S_y \end{pmatrix} \qquad (5)
$$

and

$$
\left(\delta + i\frac{\alpha}{2}\right)\begin{pmatrix} R_x-R_y \\ S_x-S_y \end{pmatrix} = \left[\begin{pmatrix} \kappa_{11}-\kappa_{2D+} & \kappa_{1D}-\kappa_{2D-} \\ \kappa_{1D}^*-\kappa_{2D-}^* & \kappa_{11}-\kappa_{2D+} \end{pmatrix} + \begin{pmatrix} i\mu & i\mu e^{i\theta_{pc}} \\ i\mu e^{-i\theta_{pc}} & i\mu \end{pmatrix}\right.
$$
$$
\left.+ \frac{1}{\sqrt{2}}\begin{pmatrix} k & 0 \\ 0 & -k \end{pmatrix}\right]\begin{pmatrix} R_x-R_y \\ S_x-S_y \end{pmatrix}, \qquad (6)
$$

where equation (5) gives the coupling between a pair of electric-field vectors $R_x + R_y$ and $S_x + S_y$, which leads to the formation of modes A and C (Supplementary Fig. 1).

The frequencies and radiation constants of modes A and C can be then derived from equation (5) as follows:

$$
\delta_{A,C} + i\frac{\alpha_{A,C}}{2} = \kappa_{11} + \kappa_{2D+} + i\mu
$$
$$
\mp \sqrt{[(\kappa_{1D}+\kappa_{2D-})+i\mu e^{i\theta_{pc}}][(\kappa_{1D}+\kappa_{2D-})^*+i\mu e^{-i\theta_{pc}}]+\left(\frac{k}{\sqrt{2}}\right)^2}
$$
$$
= \kappa_{11} + \kappa_{2D+} + i\mu \qquad (7)
$$
$$
\mp \sqrt{[(\kappa_{1D}+\kappa_{2D-})e^{-i\theta_{pc}}+i\mu][\{(\kappa_{1D}+\kappa_{2D-})e^{-i\theta_{pc}}\}^*+i\mu]+\left(\frac{k}{\sqrt{2}}\right)^2}
$$
$$
= \kappa_{11} + \kappa_{2D+} + i\mu \mp \sqrt{[R+iI+i\mu][R-iI+i\mu]+\left(\frac{k}{\sqrt{2}}\right)^2},
$$

where $\delta_A + i\alpha_A/2$ of mode A and $\delta_C + i\alpha_C/2$ of mode C correspond to the negative and positive square-root terms, respectively, and $R$ and $I$ are defined as $R \equiv \mathrm{Re}[(\kappa_{1D}+\kappa_{2D-})e^{-i\theta_{pc}}]$ and $I \equiv \mathrm{Im}[(\kappa_{1D}+\kappa_{2D-})e^{-i\theta_{pc}}]$, respectively.

## Note on the threshold gain margin $\Delta\alpha_v$

It is difficult to specify a general value of $\Delta\alpha_v$ sufficient for single-mode oscillation in PCSELs. However, we have found that increasing $\Delta\alpha_v$ by simultaneously reducing $R$ and $\mu$ contributes to the preservation of single-mode oscillation even in the presence of a non-uniform in-plane refractive index distribution borne by various physical phenomena. These findings will be reported separately.

## Estimation of the coupling coefficients $R$, $I$ and $\mu$ of the fabricated devices

To estimate the coupling coefficients $R$, $I$ and $\mu$ of the fabricated device shown in Fig. 2a, we derived the photonic band structure around the Γ point by measuring the subthreshold spontaneous emission spectra at various radiation angles (corresponding to in-plane wave numbers), whose peak emission wavelengths and line widths corresponded to the band frequencies and radiation constants, respectively. The frequencies and radiation constants of modes A and C are plotted in Fig. 2b,c. $R$, $I$ and $\mu$ were then estimated by fitting the analytical frequencies and radiation constants given by equation (1) to their measured values.

On the other hand, it was difficult to estimate the coupling coefficients of the fabricated device shown in Fig. 4a by directly measuring the photonic band structure around the Γ point due to the pre-installed lattice-constant distribution. Thus, the coupling coefficients were instead estimated by evaluating the shape of the embedded air holes.

## Design of the pre-installed lattice-constant distribution

The distribution of the lattice-constant variation $\Delta a(x,y)$, which compensates for a temperature distribution $\Delta T_{comp}(x,y)$, is determined as follows:

$$\frac{\Delta a(x,y)}{a} = \frac{\Delta T_{comp}(x,y)}{n_{eff}} \frac{dn}{dT}. \tag{8}$$

Here, $a$ is the original lattice constant, $n_{eff}$ is the effective refractive index of the photonic crystal at room temperature, and $dn/dT$ is the rate of change of refractive index with respect to temperature. Based on equation (8) and the self-consistent analysis of photon–carrier–thermal interactions (Supplementary Text Section 5), we introduced the lattice-constant distribution shown in Fig. 3e,f.

## Data availability

The data that support the plots within this paper and other findings of this study are available within this article and Supplementary Information, and they are also available from the corresponding author upon request.

## Code availability

The details of three-dimensional coupled-wave theory simulations are available in Supplementary Information, and their associated codes are available from the corresponding author upon reasonable request.

**Acknowledgements** This work was partially supported by the project of the Council for Science, Technology and Innovation; the Cross Ministerial Strategic Innovation Promotion Program; Photonics and Quantum Technology for Society 5.0 (finding agency: QST); and the Japan Society for the Promotion of Science (Grant-in-Aid for Scientific Research 22H04915).

**Author contributions** S.N. supervised the entire project. M.Y. and S.K. performed the device fabrications and the measurements with K.Izumi, M.D.Z. and K.Ishizaki. T.I. performed the theoretical analysis with M.Y. and J.G. S.K., T.I. and M.Y. performed the numerical simulation. S.N. and M.Y. discussed the results and wrote the paper with T.I. and J.G.

**Competing interests** The authors declare no competing interests.

**Additional information**
**Correspondence and requests for materials** should be addressed to Susumu Noda.
