## [Peer Review File · Nature]

Manuscript Title: High-brightness scalable continuous-wave single-mode photonic-crystal laser

Reviewer Comments & Author Rebuttals

Reviewer Reports on the Initial Version:

Referees' comments:

Referee #1 (Remarks to the Author):

This manuscript shows the power scalability of high-brightness photonic-crystal surface-emitting lasers (PCSELS) by controlling the Hermitian and non-Hermitian couplings between the oscillation modes. This work comprehensively addresses the two fatal challenges to realize high-power (>10 W) continuous-wave PCSEL with a high beam quality (or high brightness): 1) scaling of the emitting area while maintaining the single-transversal-mode operation, and 2) sustaining the single-transversal-mode operation even with the thermal effects. As an experimental demonstration, the authors successfully realized a 3-mm diameter PCSEL with a CW output power of up to 50 W and a very high brightness of $1 \text{ GW}\cdot\text{cm}^{-2}\cdot\text{sr}^{-1}$.

The presented data are physically sound. The authors systematically validate two strategies of photonic crystal design: the discrimination in the vertical radiation loss between the fundamental and high-order modes by the control of Hermitian/non-Hermitian couplings and the pre-introduction of the lattice constant distribution to compensate for the thermally-induced effect.

This work presents an essential milestone for high-power single-mode PCSELS that can potentially replace solid-state lasers and fiber lasers. Such conventional bulk lasers are consisting of a high-power but low-brightness pump laser diode. In particular, solid-state lasers regularly require the alignment of free-space components. The alignment is not needed for fiber lasers but their footprint is never as small as a single semiconductor chip. Therefore, high-power PCSELS can significantly reduce the cost and complexity of laser systems currently used in, e.g., industrial scenes. Thus, I am convinced that this work can open up a new era of high-power lasers, and the simplicity and small footprint expand the application scenes of lasers.

In the supplementary information, the authors briefly describe the limitation of edge-emitting laser diodes and VCSEL, and the further details of the analyses presented in the main manuscript. These supplementary data address the potential technical questions of future readers.

In conclusion, I recommend this manuscript be published in Nature. I point out some technical concerns as follows. These points should be addressed during the revision.

1. The authors claim that the output of their PCSELS (presented in Fig. 2 and Fig. 4) is nearly diffraction limited, but this may not be accepted in the community of solid-state lasers as the corresponding beam quality factors M^2 are larger than 2. Although nobody has clearly defined the term nearly diffraction limited, a near diffraction limited beam is conventionally considered to be an

M2 value of <1.1 . Therefore, I suggest the authors state instead the actual beam quality parameters or brightness.

2. The beam quality M2 larger than 2 indicates that a few order modes are oscillating, despite the authors designing the photonic structure with high vertical radiation loss for the first and higher order modes. The authors should comment on this concern.

3. The concept of the Hermitian/non-Hermitian control needs a bit more detail in the main text. Currently, it is hard to follow without looking at the supplementary information. At least the authors must define the modes A-D in the main text.

4. The self-consistent simulation assumed the radial symmetry of the structure, but the symmetry of the actual structure is broken by the air holes. The author should comment on the validity of radial symmetry.

Referee #2 (Remarks to the Author):

The authors reported their recent advances in high power semiconductor photonic crystal lasers with an impressive 50W single mode operation and $1 \text{ GWcm}^{-2}\text{sr}^{-1}$ from a 3 mm sized single aperture device. The work is truly a breakthrough in semiconductor diode laser technology and demonstrates experimentally the promises towards kW diode lasers with single mode and high brightness operations. While the basic theoretical strategies have been reported earlier by the authors (E.g. Ref 16), to achieve such impressive results experimentally is still a significant milestone in this laser technology. The work can be of great interest to people of different disciplines as high power lasers are used in communications, in LiDAR and sensing, in laser manufacturing, and even biomedical fields. So, such breakthroughs can attract more attention to many people from different scientific and application domains.

The work is well presented and the reviewer only has a few comments below:

(1) The authors mentioned the control of R and u to achieve large laser threshold margin between the fundamental mode and the higher order mode (Fig. 1(e)). However, it is not clear how much is needed in order to maintain the single mode operation. Such margin also limits the maximum bias current for the laser before higher order mode starts lasing. Some discussions on the specific requirement in threshold margin would be helpful.

(2) In Fig. 1(d), the authors mentioned the use of the p-DBR and the separation between DBR and cavity to control the vertical feedback (non-Hermitian coupling, also in SI Section 4). Such control is done during the epitaxial regrowth process. Will the etching profile and the regrowth profile of the photonic crystal cavity play a role here? Please elaborate.

(3) In Fig. 3(a), the authors presented the temperature rise in the cavity and proposed a lattice contact compensation. The temperature rise profile in Fig. 3(a) is specific to a bias current level and

certain heatsink design (also in SI Section 5). While single mode operation is indeed achieved around 110A, as shown in Fig. 4(e)). However multimode operation occurs at lower bias current. It could be interesting to see the range of the operation at different bias current conditions for single mode operation (e.g. from 100A to 11A and maybe even higher currents?) and whether it is possible to achieve single mode laser dynamically with heatsink designs?

Referee #3 (Remarks to the Author):

One of the important goal of laser research is to achieve sources with extremely high optical power, good beam quality and high electrical to optical conversion efficiencies. In the state of the art, this is achieved by using semiconductor lasers that pump a solid-state laser. While this arrangement is very convenient, generation of high quality and high power beams directly from a semiconductor laser device would enable even higher efficiency and a much more compact package. The work of the authors demonstrate a breakthrough in this search, by reporting a 50W laser based on a photonic crystal device where both Hermitian and non-Hermitian couplings are optimized to suppress the appearance of higher order modes.

The importance of this paper lies in the demonstration that after many decades after their first invention, photonic crystal enable devices with unique characteristics unmatched by other techniques. The paper is in general well written but could be further improved.

Firstly, the results are reported too incrementally in the text; as first the results are discussed in pulsed mode, then the effect of the temperature gradient is shown by simulations and then the final device results is demonstrated in c.w. I personally think that while chirping the periods to compensate for a given temperature profile is not trivial to implement, it is conceptually relatively simple. I would therefore relegate the content of Fig. 3 and some of the descriptive material to supplementary material and focus on the final result.

Then a few technical questions and comments:

- a) What is the nature of the emission in pulsed mode below 20A in Fig.2d? Is it spontaneous emission or thermal heating? What is the numerical aperture of the collecting optics? Since the emission is single mode spatially, the power below threshold into the laser mode should be very tiny.
- b) What is the origin of the discrepancy between the computed and measured band structure of the mode A in Fig. 2c?
- c) To which extend is the high slope efficiency arising from the nature of the mode discrimination mechanism? I believe one of the very favorable aspect of this device is that the gain discrimination between modes is achieved by increasing the vertical radiation loss of the higher transverse mode and not from lateral losses.
- d) The statement in the conclusion about extrapolating this result to a 10mm, 1kW laser from the 3mm, 50W presented in the paper should be better supported. Indeed, here the single mode operation is achieved over a limited range of currents thanks to an accurate compensation of the thermal gradients. It is not obvious to me that this remains possible for a 10x larger area device.
- e) In the supplementary material, the authors discuss another approach that was recently proposed in their reference 17. I do agree with the authors that the work presented in ref 17 raises issues concerning the optical efficiency as the device is scaled. Also, discussing the two approaches brings

benefits to the reader by highlighting how two different approach to PCSEL can be implemented, but the discussion should then be done in a clearer and more detailed manner. Indeed, the statements “the vertical emission converges to zero when the resonator diameter is widened to larger scales” is true only when assuming some finite in-plane optical losses - which, I agree is very reasonable - but was for some reason neglected in the discussion of Ref. 17 where only radiative losses are considered. These points should be discussed in a more factual and detailed manner.

Author Rebuttals to Initial Comments:

Response to the Reviewers

We are grateful to the three Reviewers for their positive evaluation of our work and their useful suggestions that have helped us to improve our paper. As indicated in the following response, we have addressed all the concerns and suggestions of the Reviewers in the revised version of our paper. Revised and newly included sentences are shown in blue in the revised manuscript.

[Reply to Reviewer #1]

General comment

This manuscript shows the power scalability of high-brightness photonic-crystal surface-emitting lasers (PCSELS) by controlling the Hermitian and non-Hermitian couplings between the oscillation modes. This work comprehensively addresses the two fatal challenges to realize high-power (>10 W) continuous-wave PCSEL with a high beam quality (or high brightness): 1) scaling of the emitting area while maintaining the single-transversal-mode operation, and 2) sustaining the single-transversal-mode operation even with the thermal effects. As an experimental demonstration, the authors successfully realized a 3-mm diameter PCSEL with a CW output power of up to 50 W and a very high brightness of $1 \text{ GW}\cdot\text{cm}^{-2}\cdot\text{sr}^{-1}$.

The presented data are physically sound. The authors systematically validate two strategies of photonic crystal design: the discrimination in the vertical radiation loss between the fundamental and high-order modes by the control of Hermitian/non-Hermitian couplings and the pre-introduction of the lattice constant distribution to compensate for the thermally-induced effect.

This work presents an essential milestone for high-power single-mode PCSELS that can potentially replace solid-state lasers and fiber lasers. Such conventional bulk lasers are consisting of a high-power but low-brightness pump laser diode. In particular, solid-state lasers regularly require the alignment of free-space components. The alignment is not needed for fiber lasers but their footprint is never as small as a single semiconductor chip. Therefore, high-power PCSELS can significantly reduce the cost and complexity of laser systems currently used in, e.g., industrial scenes. Thus, I am convinced that this work can open up a new era of high-power lasers, and the simplicity and small footprint expand the application scenes of lasers.

In the supplementary information, the authors briefly describe the limitation of edge-emitting laser diodes and VCSEL, and the further details of the analyses presented in the main manuscript. These supplementary data address the potential technical questions of future readers.

In conclusion, I recommend this manuscript be published in Nature. I point out some technical concerns as follows. These points should be addressed during the revision.

Authors' Reply

We are grateful to the Reviewer for his/her positive evaluation of our work. We are greatly encouraged by the Reviewer's comment. Each of the Reviewer's comments is addressed below.

Comment 1

The authors claim that the output of their PCSEs (presented in Fig. 2 and Fig. 4) is nearly diffraction limited, but this may not be accepted in the community of solid-state lasers as the corresponding beam quality factors M^2 are larger than 2. Although nobody has clearly defined the term nearly diffraction limited, a near diffraction limited beam is conventionally considered to be an M^2 value of <1.1 . Therefore, I suggest the authors state instead the actual beam quality parameters or brightness.

Authors' Reply

We thank the Reviewer for the useful suggestion. Indeed, as the Reviewer commented, there is no clear definition of the term “nearly-diffraction-limited”, and different communities may have different impressions of the term. We would like to address this concern as follows: In the manuscripts, we remove all uses of the term “nearly-diffraction-limited”, and we mention the values of brightness or M^2 as appropriate.

Comment 2

The beam quality M^2 larger than 2 indicates that a few order modes are oscillating, despite the authors designing the photonic structure with high vertical radiation loss for the first and higher order modes. The authors should comment on this concern.

Authors' Reply

We thank the Reviewer for this comment. In our device, the value of the beam quality factor, M^2 , can be larger than 1 even for single-mode oscillation because the in-plane electromagnetic field intensity profile is deviated from an ideal Gaussian profile (more specifically, the intensity profile more closely resembles a super-Gaussian). For example, the intensity profile I with a super-Gaussian of an order n is given by the following equation:

$$I = I_0 \exp\left(-2\left(\frac{r}{w_0}\right)^{2n}\right)$$

where I_0 is the peak intensity, r is the radial coordinate, and w_0 is the beam width. The intensity profile is shown for several n in Fig. R1(a). M^2 of the super-Gaussian beam increases with the order n , as

Fig. R1 | Dependence of the beam quality factor M^2 on the super-Gaussian order n . (a) Beam profiles for the super-Gaussian order $n = 1$ (corresponding to a Gaussian profile), 3, 5, and 10. (b) M^2 as a function of the super-Gaussian order n (referred to [R1]).

shown in Fig. R1(b) [R1].

The reasons why the intensity profile of our device resembles a super-Gaussian even in the case of oscillation in a single, fundamental mode are as follows: The fundamental mode originally has a nearly Gaussian profile as shown in Fig. 1a in the main text. However, the current injection distribution is designed to have a uniform or flat profile as described in the main text. When injected current is increased, the electric field distribution tends to deform into the shape close to the current injection distribution due to mutual interactions between photons and carriers (e.g., spatial hole burning effects and associated changes in the in-plane refractive index distribution). As a result, the intensity profile approaches a super-Gaussian profile. (In future work, we would like to try to control the current injection distribution so that the intensity profile is not deformed and $M^2 \sim 1$ can be realized.) According to the above discussions, we have added the following sentences to the revised manuscript.

Lines 243–245 in the revised manuscript:

“We note that $M^2 \geq 2$ in spite of single-mode oscillation due to the super-Gaussian electromagnetic-field intensity profile caused predominantly by the uniform current injection.”

[R1] Duarte, F. J. Laser pulse phenomena and applications. *InTech*, <http://doi.org/10.5772/881> (2010).

Comment 3

The concept of the Hermitian/non-Hermitian control needs a bit more detail in the main text. Currently, it is hard to follow without looking at the supplementary information. At least the authors must define the modes A-D in the main text.

Authors' Reply

We thank the Reviewer for this useful suggestion. We had to create the space necessary to focus on the important experimental results by omitting the details about Hermitian/non-Hermitian control from the manuscript. However, following the Reviewer's suggestion, we have included more details about Hermitian/non-Hermitian control in Supplementary Section 3, and we have included the definitions of modes A-D in the revised version of the manuscript as follows.

Lines 111–116 in the revised manuscript:

“Due to such optical couplings among the four fundamental waves, four resonant cavity modes are constructed. In a double-lattice photonic crystal with mirror symmetry about the axis of $y=x$, two of these modes (labelled A and C) are anti-symmetric modes, whose electric-field vectors are anti-symmetric about this axis, and the remaining two modes (labelled B and D) are symmetric modes, whose electric-field vectors are symmetric about this axis.”

Comment 4

The self-consistent simulation assumed the radial symmetry of the structure, but the symmetry of the actual structure is broken by the air holes. The author should comment on the validity of radial symmetry.

Authors' Reply

We thank the Reviewer for this comment. As the Reviewer has pointed out, the air-hole structures are asymmetric. Such asymmetry has been properly taken into account in our self-consistent analysis. (i.e., we do not assume radial symmetry in our analysis). To make this point clearer, we have explicitly indicated all dependencies on x and y in Eq. (2) as follows.

$$\frac{\Delta a(x,y)}{a} = \frac{\Delta T_{\text{comp}}(x,y)}{n_{\text{eff}}} \frac{dn}{dT}. \quad (2)$$

[Reply to Reviewer #2]

General comment

The authors reported their recent advances in high power semiconductor photonic crystal lasers with an impressive 50W single mode operation and $1 \text{ GWcm}^{-2}\text{sr}^{-1}$ from a 3 mm sized single aperture device. The work is truly a breakthrough in semiconductor diode laser technology and demonstrates experimentally the promises towards kW diode lasers with single mode and high brightness operations. While the basic theoretical strategies have been reported earlier by the authors (E.g. Ref 16), to achieve such impressive results experimentally is still a significant milestone in this laser technology. The work can be of great interest to people of different disciplines as high power lasers are used in communications, in LiDAR and sensing, in laser manufacturing, and even biomedical fields. So, such breakthroughs can attract more attention to many people from different scientific and application domains.

Authors' Reply

We are grateful to the Reviewer for his/her positive evaluation of our work. We are greatly encouraged by the Reviewer's comments. Each of the Reviewer's comments is addressed below.

Comment 1

The authors mentioned the control of R and u to achieve large laser threshold margin between the fundamental mode and the higher order mode (Fig. 1(e)). However, it is not clear how much is needed in order to maintain the single mode operation. Such margin also limits the maximum bias current for the laser before higher order mode starts lasing. Some discussions on the specific requirement in threshold margin would be helpful.

Authors' Reply

We thank the Reviewer for this important comment. Honestly speaking, it is difficult to specify a value of $\Delta\alpha_v$ that is sufficient for single-mode oscillation in actual devices because this value is affected by many physical phenomena, which affect the in-plane frequency distribution, as discussed in this paper.

[This has been redacted]

[This has been redacted]

Nevertheless, we may say that $\Delta\alpha_v$ required for single-mode oscillation can be decreased by making the frequency distribution uniform. We plan to investigate the relationship between $\Delta\alpha_v$ and single-mode discrimination in more detail, including the effects of various variations of the distribution, and to report these results in a future paper. In consideration of the above discussion, we have added the following sentences to the revised manuscript.

Lines 143–148 in the revised manuscript:

“Note that although it is difficult to specify a general value of $\Delta\alpha_v$ sufficient for single-mode oscillation, we have found that increasing $\Delta\alpha_v$ by simultaneously reducing R and μ contributes to the preservation of single-mode oscillation even in the presence of a non-uniform in-plane refractive index distribution borne by various physical phenomena. These findings will be reported separately.”

Comment 2

In Fig. 1(d), the authors mentioned the use of the p-DBR and the separation between DBR and cavity to control the vertical feedback (non-Hermitian coupling, also in SI Section 4). Such control is done during the epitaxial regrowth process. Will the etching profile and the regrowth profile of the photonic crystal cavity play a role here? Please elaborate.

Authors' Reply

We thank the Reviewer for this comment. Although the etching and regrowth profiles of the air holes do indeed influence non-Hermitian coupling, the etching and regrowth conditions were kept constant throughout our experiments, so these profiles, and thus their influence on non-Hermitian coupling, were mostly the same across all fabricated samples. Nevertheless, if the depth of the air holes is slightly changed in fabrication, then we can adjust thickness of the p-cladding layer commensurately, so that the vertical feedback between the DBR and the cavity remains constant.

Comment 3

(3) In Fig. 3(a), the authors presented the temperature rise in the cavity and proposed a lattice constant compensation. The temperature rise profile in Fig. 3(a) is specific to a bias current level and certain heatsink design (also in SI Section 5). While single mode operation is indeed achieved around 110A, as shown in Fig. 4(e)). However multimode operation occurs at lower bias current.

It could be interesting to see the range of the operation at different bias current conditions for single mode operation (e.g. from 100A to 110A and maybe even higher currents?) and whether it is possible to achieve single mode laser dynamically with heatsink designs?

Authors' Reply

We thank the Reviewer for this important comment. As the Reviewer pointed out, oscillation in a few modes was observed at injection currents below 100 A because the temperature distribution did not match the pre-installed distribution in this range. We believe that it could be possible to achieve single-mode oscillation over a wider range of injection currents with heatsink design as the Reviewer has suggested.

[This has been redacted]

[Reply to Reviewer #3]

General comment

One of the important goal of laser research is to achieve sources with extremely high optical power, good beam quality and high electrical to optical conversion efficiencies. In the state of the art, this is achieved by using semiconductor lasers that pump a solid-state laser. While this arrangement is very convenient, generation of high quality and high power beams directly from a semiconductor laser device would enable even higher efficiency and a much more compact package. The work of the authors demonstrate a breakthrough in this search, by reporting a 50W laser based on a photonic crystal device where both Hermitian and non-Hermitian couplings are optimized to suppress the appearance of higher order modes.

The importance of this paper lies in the demonstration that after many decades after their first invention, photonic crystal enable devices with unique characteristics unmatched by other techniques. The paper is in general well written but could be further improved.

Authors' Reply

We are grateful to the Reviewer for his/her positive evaluation of our work. We are greatly encouraged by the Reviewer's comment. Each of the Reviewer's comments is addressed below.

Comment 1

Firstly, the results are reported too incrementally in the text; as first the results are discussed in pulsed mode, then the effect of the temperature gradient is shown by simulations and then the final device results is demonstrated in c.w. I personally think that while chirping the periods to compensate for a given temperature profile is not trivial to implement, it is conceptually relatively simple. I would therefore relegate the content of Fig. 3 and some of the descriptive material to supplementary material and focus on the final result.

Authors' Reply

We thank the Reviewer for this important suggestion. Although the introduction of a lattice-constant distribution to compensate for the temperature distribution seems to be conceptually simple at one glance, it is actually complex because of the need to consider the multi-physics action of photons, carriers, and thermal effects, as mentioned in the main text. For example, introducing a lattice-constant distribution to compensate a particular temperature distribution also changes the distributions of carrier consumption and heat density, which, in turn, change the temperature distribution. Even in this complex multi-physics situation, we successfully demonstrate this compensation. Thus, we consider that an inclusion of the above analytical results as Fig. 3 in the main manuscript and a step-by-step

presentation would be important and useful for readers especially from different disciplines. We would like to request the Reviewer's kind understanding on this matter.

Comment 2

What is the nature of the emission in pulsed mode below 20A in Fig.2d? Is it spontaneous emission or thermal heating? What is the numerical aperture of the collecting optics? Since the emission is single mode spatially, the power below threshold into the laser mode should be very tiny.

Authors' Reply

We thank the Reviewer for this comment. The output power below 20 A shown in the I-L characteristic in Fig. 2d is due to the spontaneous emission of light. In this measurement, the detector was placed close to the optical facet of our device, so the detector collected light emitted spontaneously in multiple modes across a wide range of angles (corresponding to a numerical aperture of 0.25~0.3), including modes other than the final oscillating laser mode.

Comment 3

What is the origin of the discrepancy between the computed and measured band structure of the mode A in Fig. 2c?

Authors' Reply

We thank the Reviewer for this comment. We attribute the discrepancies between experiments and measurements to the effects of measurement uncertainty. The measured radiation coefficients in Fig. 2c are obtained from the full width at half maximum of the spontaneous emission spectrum corresponding to the resonant mode. These measurements contain two main sources of uncertainty: The first is the measurement accuracy of the spectrometer we used, which resulted in an uncertainty of about ± 0.05 nm in the measured linewidth (corresponding to ± 12 cm^{-1} in terms of the radiation coefficient). The second is the positional accuracy of the rotary stage used to fix the measurement angle, which resulted in an uncertainty of ± 0.005 degrees in the measurement angle (corresponding to $\pm 2.58 \times 10^{-5} \times 2\pi/a$ in terms of the wavenumber). Nevertheless, we believe that the theoretical curve is a reasonably good fit to the experimental measurements of the radiation constants and the frequencies, and that the coupling coefficients derived from this curve are reasonable. Considering the above discussion, we have revised Fig. 2c in the revised manuscript by adding error bars to the measured data points.

Comment 4

To which extent is the high slope efficiency arising from the nature of the mode discrimination mechanism? I believe one of the very favorable aspect of this device is that the gain discrimination between modes is achieved by increasing the vertical radiation loss of the higher transverse mode and not from lateral losses.

Authors' Reply

We thank the Reviewer for this comment. The Reviewer's comment is correct. Indeed, we have achieved mode discrimination by increasing the vertical radiation loss of the higher-order transverse modes. In this way, we can maintain mode discrimination even when the in-plane loss is decreased, which is favorable for increasing the slope efficiency. This slope efficiency can be even further increased by reducing the absorptive losses inside of the material. We would like to focus on reducing the absorptive losses in future work.

Comment 5

The statement in the conclusion about extrapolating this result to a 10mm, 1kW laser from the 3mm, 50W presented in the paper should be better supported. Indeed, here the single mode operation is achieved over a limited range of currents thanks to an accurate compensation of the thermal gradients. It is not obvious to me that this remains possible for a 10x larger area device.

Authors' Reply

We thank the Reviewer for this comment. We believe that the concept presented in the current work is basically applicable to 10-mm devices. However, as the Reviewer has surmised, to do so might require even more precise control of the in-plane band-edge frequency distribution. In addition, in order to compensate the temperature distribution, additional concepts [This has been redacted] might also be necessary. Out of consideration of such possibilities, we have revised statements about 10-mm devices while avoiding use of the word “straightforward” as follows.

Lines 83–85 in the revised manuscript:

“The strategies demonstrated here are expected to be applicable to scaling up the diameter of the device to 10 mm, leading to 1-kW-class, high-beam-quality operation of PCSELs.”

Lines 260–263 in the revised manuscript:

“Controlling the Hermitian and non-Hermitian coupling coefficients (R , I , and μ) and introducing a lattice-constant distribution suitable for devices of even larger scales (e.g., 10-mm diameters) is

expected to contribute to the realization of 1-kW-class, high-beam-quality operation of PCSELS.”

Comment 6

In the supplementary material, the authors discuss another approach that was recently proposed in their reference 17. I do agree with the authors that the work presented in ref 17 raises issues concerning the optical efficiency as the device is scaled. Also, discussing the two approaches brings benefits to the reader by highlighting how two different approach to PCSEL can be implemented, but the discussion should then be done in a clearer and more detailed manner. Indeed, the statements “the vertical emission converges to zero when the resonator diameter is widened to larger scales” is true only when assuming some finite in-plane optical losses - which, I agree is very reasonable - but was for some reason neglected in the discussion of Ref. 17 where only radiative losses are considered. These points should be discussed in a more factual and detailed manner.

Authors' Reply

We thank the Reviewer for this suggestion. Following the Reviewer's suggestion, we made a more detailed comparison between the work presented in Ref. 17 and our work as described below. The lasers based on an open-Dirac singularity of Ref. 17 feature triangular-lattice photonic-crystal structures with symmetric circular lattice points which are carefully tuned to realize the open-Dirac singularity. Since this structure possesses C_{6v} symmetry, the vertical radiation constant α_v of the lasing mode converges to zero when the resonator size is widened to larger scales. In addition to α_v , in actual semiconductor lasers, a fixed amount of fundamental material absorption loss α_0 (typically $\geq 1 \text{ cm}^{-1}$) exists due to free-carrier absorption in the cladding layers, etc. Considering these facts, the slope efficiency (i.e., surface emission efficiency) of the laser, which is proportional to $\alpha_v/(\alpha_v+\alpha_{//}+\alpha_0)$ where $\alpha_{//}$ denotes the in-plane loss, inevitably converges to zero at larger scales. The random scattering of light induced by fabrication disorders may recover the slope efficiency to some extent [R2] while sacrificing the beam quality. In addition, owing to the C_{6v} symmetry, α_v of modes on different band edges also converge toward zero as the device size increases, so competition between the modes of different band edges might hinder single-mode operation. It should be also noted that, in Ref. 17, rigorous simulations of I - L characteristics considering carrier-photon interactions were not performed, and experimental demonstrations were limited to resonator (emission) sizes of $64 \mu\text{m}$ under pulsed optical pumping (not under CW current injection).

On the other hand, in our approach based on Hermitian/non-Hermitian control in a double-lattice PCSEL, which has no rotational symmetry, the radiation constant of the lasing band-edge mode (mode A) can be controlled to an appropriate value while keeping those of the other band-edge modes (modes B, C, and D) much higher [R3], even at large scales. In this way, lasing in a single mode can be

obtained while also ensuring a high slope efficiency and a high-beam-quality single-lobed beam pattern.

A detailed comparison between the approach based on an open-Dirac singularity and the approach based on Hermitian/non-Hermitian control can be summarized in Table R1 below.

Following the Reviewer's suggestion, we have added the above discussion to the revised version of Supplementary Section 2. We believe this discussion will provide useful information to help readers distinguish the individual features of each work.

[R2] Private communication with the authors of reference [17].

[R3] Inoue, T. et al. General recipe to realize photonic-crystal surface-emitting lasers with 100-W-to-1-kW single-mode operation. *Nat. Commun.* **13**, 3262 (2022).

Table R1| Comparison between PCSELs based on an open-Dirac singularity and those based on Hermitian/non-Hermitian control

	PCSELs based on open-Dirac singularity (Ref.17 in the main text)	PCSELs based on non-Hermitian/Hermitian control (this work and [R3])
Structure		
Periodicity	Triangular-lattice	Square-lattice (Double-lattice)
Unit cell and symmetry	Single circular hole (C_{6v} symmetry)	Elliptic and circular holes (Mirror symmetry along $y=x$, but no rotational symmetry)
Cladding	Air	Semiconductor
Basic properties		
Radiation constant of the lasing band-edge mode	Converges to zero for large devices (C_{6v} symmetry)	Arbitrarily tunable (no rotational symmetry)
Slope efficiency (considering material absorption loss)	Converges to zero when the size increases and disorders do not exist.	Can be kept high
Beam shape	Multi-lobe for small size, and random beam for large size	Single-lobed
Simulation		
Methods	3D finite-element method	3D coupled-wave theory

Device size	Up to 130 μm	3-10 mm
Carrier effect	Not considered	Considered
Lasing spectra	Not shown	Single mode
I-L characteristics	Not shown	50-100 W for 3 mm 500-1000 W for 10 mm
Experiment		
Device size	Up to 65 μm	3 mm
Excitation	Optical pumping	Electrical injection
Operation	Pulsed	CW
Output power	Not shown	CW 50W
Beam shape	Multi-lobed for small size (Up to 65 μm)	Single-lobed
Brightness	Not shown	1 $\text{GWcm}^{-2}\text{sr}^{-1}$ (rivals bulky lasers)

Reviewer Reports on the First Revision:

Referees' comments:

Referee #1 (Remarks to the Author):

During the revision, the authors convincingly answered all the concerns raised by the reviewers and accordingly improved the manuscript. The output of their PCSELS is indeed in single mode, but the M2 factor is not equal to unity because of the non-gaussian beam shape. The correction applied by the authors, that the output was single-mode but its super-Gaussian field distribution makes M2 larger than 2, should be helpful for future readers.

Again, this work is an essential milestone for high-power single-mode PCSELS for the replacement of solid-state lasers and fiber lasers. This is indeed a breakthrough in the field of laser technology. In conclusion, I recommend this manuscript be published in the Nature.

Referee #2 (Remarks to the Author):

Recommends publication

Referee #3 (Remarks to the Author):

The authors have answered very well the questions and clarifications asked by the referees. I recommend rapid publication of the manuscript.